# “Tear-Drop Appearance” Wrap: A Novel Implant Coverage Method for Creating Natural Contour in Prepectoral Prosthetic-Based Breast Reconstruction

**DOI:** 10.3390/jcm11154592

**Published:** 2022-08-05

**Authors:** Hong-il Kim, Byeong-seok Kim, Jin-hyung Park, Hyung-suk Yi, Hyo-young Kim, Jin-hyuk Choi, Sung-ui Jung, Yoon-soo Kim

**Affiliations:** 1Department of Plastic and Reconstructive Surgery, Kosin University Gospel Hospital, Kosin University College of Medicine, Busan 49267, Korea; 2Department of Surgery, Kosin University Gospel Hospital, Kosin University College of Medicine, Busan 49267, Korea

**Keywords:** breast cancer, breast reconstruction, mammaplasty, mastectomy, acellular dermal matrix

## Abstract

Various implant wrapping methods with acellular dermal matrix (ADM) have been introduced, but most focus on random trimming and suturing aimed to maximize implant coverage. Here we present our clinical experience using a “tear-drop appearance” wrapping method to achieve natural contours through upper pole volume replacement. We retrospectively reviewed the data of 56 consecutive cases of prepectoral prosthetic-based breast reconstruction (PPBR) using this wrapping method following nipple-sparing mastectomy between March 2020 and June 2021. The “tear-drop appearance” wrapping design creates an anatomical tear-drop–shaped pocket to encourage lower pole fullness and create a natural contour through upper pole volume replacement by ADM. Patients’ baseline characteristics, operative data, and complications were analyzed. Aesthetic outcomes were measured using the BREAST-Q and Aesthetic Item Scale (AIS). A successful reconstruction was achieved without major complications and using a single ADM sheet. Four types and three sizes of ADMs were used. The mean resected breast tissue weight was 274.3 g, while the mean implant volume was 230.0 cc. The average BREAST-Q and AIS scores were 4.6 ± 0.8 and 4.5 ± 0.7, respectively. Owing to its simplicity, reproducibility, and effectivity, this method is an excellent implant coverage option that achieves a natural contour in PPBR.

## 1. Introduction

Prosthetic-based breast reconstruction is the most widely used option after mastectomy in Korea and the United States [1,2]. The most widely used prosthesis position over the past few decades had been the subpectoral or dual-plane [3]. Prepectoral prosthetic-based breast reconstruction (PPBR) was introduced by Snyderman and Guthrie in the 1970s, but it was difficult to be widely used due to mastectomy skin flap necrosis, implant extrusion, etc. [4,5]. With the development of mastectomy surgical techniques and the use of acellular dermal matrix (ADM), PPBR began to gradually increase [6]. PPBR is a viable option for immediate breast reconstruction after mastectomy [7]. As PPBR gains notoriety and acceptance as an alternative to subpectoral or dual-plane reconstruction, adjustments will enable a more optimal result in terms of aesthetics and patient satisfaction [6,7]. The disadvantages of subpectoral and dual-plane have been reported to be animation deformity, pain, muscle spasm, and implant malposition caused by contracture of the pectoralis muscle [8]. However, PPBR may reduce postoperative pain and the risk of postoperative animation deformity by preserving the pectoralis major muscle [9].

Acellular dermal matrix (ADM) was introduced in 2006 and is now widely used for PPBR, allowing implant coverage and support [10,11,12]. The use of ADM in PPBR has several advantages. Firstly, it leads to better aesthetic results by providing a scaffold for implant positioning and soft tissue support, allowing the surgeon to control the breast pocket [7]. Secondly, ADM decreases implant rippling and capsular contracture rates by offering an additional layer of coverage between the implant and the mastectomy flap [6].

Various methods for using ADM in a PPBR have been published [13,14,15]. However, previously published reports on wrapping techniques of prosthesis coverage revealed two potential drawbacks to its use: first, a lack of standardization; and second, a conspicuous and palpable superior prosthesis edge [9].

Thus, here we propose the “tear-drop appearance” wrapping method as a novel implant coverage technique that creates an anatomical tear-drop-shaped pocket. The use of this technique in PPBR encourages lower pole fullness and creates a natural contour through upper pole volume replacement by ADM without a palpable superior implant edge.

## 2. Materials and Methods

### 2.1. Patients Selection and Study Design

We retrospectively reviewed the cases of all patients who underwent nipple-sparing mastectomy (NSM) combined with immediate PBBR at our institution between March 2020 and March 2021 performed by two plastic surgeons (J.-h.P and Y.-s.K). Exclusion criteria were having undergone radiotherapy (RTx.) before or after surgery or having undergone bilateral breast surgery. Patient demographics, body mass index, resected breast tissue weight during surgery, comorbidities, complications, and reconstruction details were recorded and analyzed. MENTOR^®^ MemoryGel™ breast implants (Mentor, Santa Barbara, CA, USA) were used for the breast prosthesis and Megaderm^®^ (L&C Bio, Seoul, Korea), CGDerm^®^ (CGBIO Inc., Seongnam, Korea), Bellacell^®^ (Hans Biomed, Seoul, Korea), or DermACELL^®^ (LifeNet Health, Virginia Beach, VA, USA) sized 16 × 16, 18 × 18, or 20 × 20 cm^2^ were used for the ADM. Clinical photographs were obtained before and 21 days, 3 months, and 12 months after surgery.

All study participants provided written informed consent to store their medical information in the database and use it for research purposes. The study protocol was approved by the Institutional Review Board of Kosin University Gospel Hospital of Korea (KUGH 2022-03-041). All procedures were performed in accordance with the ethical standards of the Institutional and National Research Committee and the 1964 Helsinki Declaration and its later amendments.

### 2.2. Surgical Procedures and ADM Warpping Technique

In most cases, NSM is performed using an inframammary fold (IMF) incision. Immediately after the NSM was completed by the breast surgeons (J.-h.C, S.-u.J), perfusion of the mastectomy skin flaps was assessed using an infrared camera (Fluobeam^®^, FLUOPTICS, Grenoble, Auvergne-Rhône-Alpes, France) using an intravenous injection of 5 mL of indocyanine green dye. The mastectomy flaps showed adequate perfusion and homogeneous thickness (intact subcutaneous fat without dermal exposure) and the PBBR was immediately performed. ADM type was selected according to mastectomy flap thickness, while implant volume was selected according to breast base width.

To prevent seroma formation, fenetration was performed by the creation of several stab incisions using a no. 11 surgical scalpel blade. The basement membrane side of the ADM was spread out on top, and both sides and the bottom were divided into thirds (Figure 1A and Figure 2A). The implant was placed in the center of the ADM with the back side facing upward, and extension lines were drawn across the width of the implant to the top, bottom, and both sides. A triangle was drawn by connecting the lower one third point and the point where the extension line of the width met. The upper part marked the lateral one half of the implant width extension line and connected it to the area where the width extension line met to form a triangle. First, parts of the triangles in the design were cut and kept (Figure 1B and Figure 2B). The lower pole pocket was formed using Vicryl^®^ 3-0 sutures (Ethicon Inc., Somerville, NJ, USA) (Figure 2C), and the middle sides were folded and fixed at the lower pockets (Figure 2D). Both lateral upper sides were gathered at the center and fixed (Figure 2E). Two of the cut triangles made from the bottom were sutured to the area not covered by the ADM, and the two made from the top were fixed to the upper ADM to create upper pole volume replacement (Figure 2F,I).

The breast pocket was irrigated first with a betadine solution and then irrigated with an antibiotic (gentamicin) and warm normal saline before implant placement. The implant was placed on the mastectomy skin flap and three anchoring points marked on the breast upper pole margin drawn before surgery and upper ADM margin. Before inserting the ADM-wrapped implant, pullout sutures were made using Prolene^®^ (Ethicon Inc.) 2-0 sutures outside the pocket and the implant was inserted by pulling it upwards (Figure 3A). The marked points were anchored with bolsters (Figure 2H and Figure 3B). A Jackson–Pratt (JP) drain was placed behind the implant, and the wound was closed in layers with Vicryl^®^ 3-0, 5-0 sutures and Mersilk^®^ (Ethicon Inc.) 6-0 sutures. A JP drain was removed if the daily drainage was <30 mL of serum for 2 consecutive days, and the patient was discharged. Total stitching off was performed on the 10th postoperative day, while the upper pole bolster was removed on the 21st postoperative day.

### 2.3. Satisfaction Survey

We evaluated patient-reported outcomes using the BREAST-Q questionnaire [15] using reconstruction module version 2.0 for self-reported satisfaction regarding reconstructed breasts and implants. To assess aesthetic outcomes, five board-certified plastic surgeons (J.-h.P., Y.-s.K., H.-i.K., H.-s.Y., and H.-y.K.) blinded to the reconstructive approach evaluated the photographs obtained before and at least 12 months after surgery. The Aesthetic Items Scale (AIS) was used to assess and score aesthetic outcomes after the breast reconstruction [16]. Each plastic surgeon was asked to assess standardized photographs of patients obtained from different angles (frontal view, at a 45-degree angle, and lateral view (pre- and postoperatively)). The breasts were evaluated for volume, shape, symmetry, scarring, and nipple–areola complex [16]. A five-point Likert scale was used for each item, corresponding to “very dissatisfied”, “dissatisfied”, “neutral”, “satisfied”, and “very satisfied” [16]. Data were analyzed using Microsoft Excel version 16.53 (Microsoft Corp., Redmond, WA, USA).

## 3. Results

Between March 2020 and March 2021, 73 patients underwent NSM plus PPBR. A total of 56 patients were included in the study; those who received RTx. or underwent bilateral surgery were excluded. The mean patient age was 50.2 years, while the mean body mass index was 24.0. The most common breast shape was conical, with 27 patients (48.2%) having ptotic breasts (Table 1). The mean breast base width was 13.4 cm and the mean JP drain insertion period was 10.4 days. The mean resected breast tissue weight was 274.3 g, while the mean implant volume was 230.0 cc. The ADM use rates were as follows: Megaderm^®^, 26 patients (46.4%); CGDerm^®^, 18 patients (32.1%); Bellacell^®^, 9 patients (16.1%); and DermACELL^®^, and 3 patients (5.4%) (Table 2).

Two (3.6%) patients developed postoperative complications during follow-up (Table 1). Seroma and hematoma, observed in one patient each, were evaluated using physical examination and ultrasonography. Seroma improved with several aspirations, while hematoma liquefied after two subcutaneous injections of hyaluronidase (1.5 mL) at 1-week intervals. No volume change or rippling of the breasts was observed during follow-up, and no seroma or hematoma were detected on computed tomography and ultrasonography performed at 6 and 12 months postoperatively. No major complications required reoperation or hospitalization (Figure 4 and Figure 5).

The average patient self-reported score on the BREAST-Q questionnaire at the 12-month follow-up was 82.6. The aesthetic scores measured at least 12 months postoperatively as reported by the five blinded plastic surgeons are listed in Table 3.

## 4. Discussion

Here, we introduced and standardized the tear-drop appearance wrapping method for PPBR in which a natural contour is created through upper pole volume replacement using ADM and lower pole fullness is encouraged, especially in situations in which it is difficult to use anatomical shaped implants due to breast implant–associated diffuse large B-cell lymphoma [17]. PPBR is currently evolving, with increasing evidence pointing toward its safety, reliability, and perceived benefits over other approaches [18,19]. The use of ADM in PPBR is inevitable, and the wrapping method proposed here can achieve a relatively feasible and cosmetically satisfactory result [10].

During the study period, there were a total of 112 cases of breast reconstruction after mastectomy. Among them, there were 73 cases where immediate PBBR was performed, and all of them used this wrapping method. Autologous breast reconstruction was performed in 33 cases with a latissimus dorsi musculocutaneous (LD) flap in 18 cases, pedicled transverse rectus abdominis myocutaneous flap in 11 cases, and deep inferior epigastric artery perforator flap in 4 cases. In the remaining 6 cases, LD flap and implant were used together. The reconstruction method was determined according to patient preference and mastectomy technique. PBBR was performed for all nipple-sparing mastectomies and autologous breast reconstruction was performed for skin-sparing mastectomies. The history of RTx. acted as an exclusion criterion in the study, because it could act as a variable such as breast atrophy and an increase in revision rate before and after RTx. [20,21]. Therefore, the surgical advantage of our novel wrapping method could be affected by the shape deformation caused by RTx., which could make accurate evaluation difficult, so it was excluded from the study. Therefore, the study population was 56.

As the survival rate of breast cancer patients increases, interest in quality of life increases as well [22]. Therefore, many patients desire a natural appearance and good cosmetic results of post-mastectomy reconstruction. As the mastectomy flap elevation technique increases, it becomes oncologically safe, preserves the nipple–areolar complex, and maintains even flap thickness, thereby improving patient satisfaction [23,24]. Our study implemented both patient self-evaluation (BREAST-Q) and professional evaluation by blinded plastic surgeons (AIS) to increase its reliability [15,16]. In the survey conducted at the 12-month follow-up, the mean BREAST-Q score was 82.6 points, indicating satisfactory results. This is a high result compared to 64.0 points in surgical satisfaction in patients who underwent prosthetic-based breast reconstruction in Pusic et al. [25]. In addition, the mean total AIS score was 21.0 points; considering that the score is out by 25 points, it represents 80% of satisfaction. This is similar to Hu et al.’s prosthetic-based breast reconstruction with an 82% aesthetic satisfaction within 5 years [26]. In particular, in breast volume and shape, high scores were recorded with 4.4 ± 0.6 and 4.5 ± 0.6 points, respectively, suggesting that the anatomical shape and lower pole fullness intended by the authors were well implemented.

Hammond et al. reported that the superior prosthesis edge is conspicuous and palpable as a drawback of PPBR [9]. However, we were able to create the natural contour of the upper pole through volume replacement using the ADM triangle fragment created while making the lower pole through the tear-drop appearance wrapping method. In addition, this wrapping technique implements lower pole fullness through minimal ADM overlap. Augmenting PPBR using ADM can be expensive given the additional cost of ADM [13]. By design, the tear-drop appearance wrapping technique minimizes waste from each individual ADM sheet, therefore enabling the implementation of the upper pole natural contour and lower pole fullness without additional ADM consumption for one implant and subsequently reducing overall procedural costs. ADM wrapping is performed on the back table, and the first assistant can simultaneously perform bleeding control and antibiotic irrigation, which has the advantages of reducing the operation time and being easy to learn with a shallow learning curve. A single sheet is used, and a stab incision is applied to reduce the occurrence of seroma and enable the use of a wider range of prostheses [27,28]. After wrapping of the prosthesis with the ADM using the tear-drop appearance wrapping method, a pullout suture is performed atop the ADM upon its insertion into the breast pocket. Thus, the effect of helping the prosthesis being placed in the position intended by the surgeon was expected. There were no cases of implant malpositioning, which demonstrates that the pullout suture effectively enforced the implant positioning.

ADM is produced from donor human cadaveric skin by decellularization and removal of the epidermis. Since ADM contains collagen, elastin, proteoglycans, laminin, and a basement membrane, it is of great help in accelerating wound healing [29]. Additionally, ADM does not induce an immune response and is widely used in reconstructive surgery [29,30]. Especially in breast surgery, the ADM reduces complications, such as capsular contracture during implant reconstruction after total mastectomy, and is also used for volume replacement in oncoplastic surgery [10,11,31]. Although the authors used four types (Megaderm^®^, CGDerm^®^, Bellacell^®^, DermACELL^®^) and three sizes (16 × 16, 18 × 18, or 20 × 20 cm^2^), good results were obtained regardless of the type and size of ADM. Therefore, this method can be applied regardless of the type and size of ADM.

Since implant-based breast reconstruction accounts for most cases in the United States and Korea, the availability of a safe and reproducible surgical technique is important [1,2]. PPBR is the next step, as compared with a submuscular approach, it allows for decreased operative time, decreased postoperative pain, more rapid recovery, a natural breast shape, and reduction in animation deformities [32]. Advances in surgical techniques have led to enhanced cosmetic results and improved psychosocial well-being of breast cancer patients [33]. Surgical complications associated with these procedures have significant outcome and cost-related implications [34,35]. Among 73 patients, two minor complications occurred but no major complications requiring reoperation. Furthermore, because we used anchoring sutures on the upper pole, the implant position was approximately fixed and implant malposition and rotation were prevented. The authors’ method will be helpful if the implant falls to the lateral side of the breast postoperatively or a double-bubble deformity occurs.

The limitations of our study include its short follow-up period and limited ability to create a ptotic breast shape. No capsular contracture or implant infections occurred in patients currently hospitalized until the 21-month follow-up. In the future, it will be necessary to identify a method of securing a longer follow-up period and an effective upper pole contour, even in ptotic breasts. Further studies of larger numbers of patients with longer follow-up periods are necessary to confirm these findings and identify commonly reported long-term complications of PPBR.

## 5. Conclusions

PPBR techniques are being developed and widely used daily. With their universal use, efforts should be made to improve their cosmetic results and patient quality of life. By using this method, the natural contour of the upper pole and the fullness of the lower pole were implemented to make the breast with an anatomical shape as much as possible. Here, we achieved successful results using an easily reproducible method with few complications and high satisfaction rates.

## Figures and Tables

**Figure 1 jcm-11-04592-f001:**
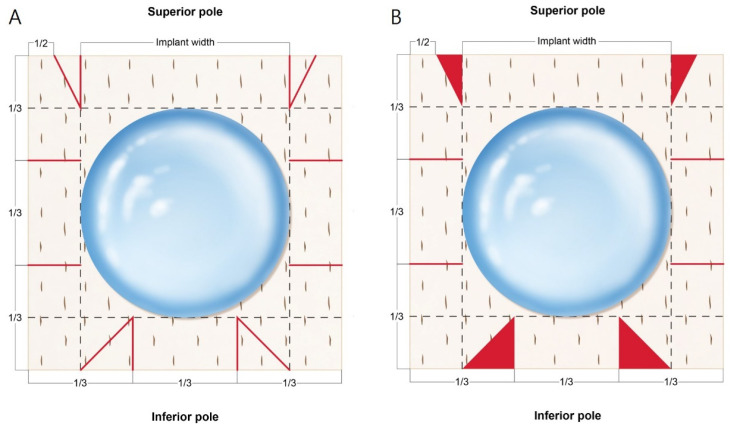
”Tear-drop appearance” wrapping technique design for breast implant. (**A**) The sides and inferior poles were divided into thirds. The dotted line indicates the implant width and the red solid line indicates the part cut with the no. 15 surgical scalpel blade. (**B**) This is a template of the implant placed after cutting. The cut triangular part (red) is retained.

**Figure 2 jcm-11-04592-f002:**
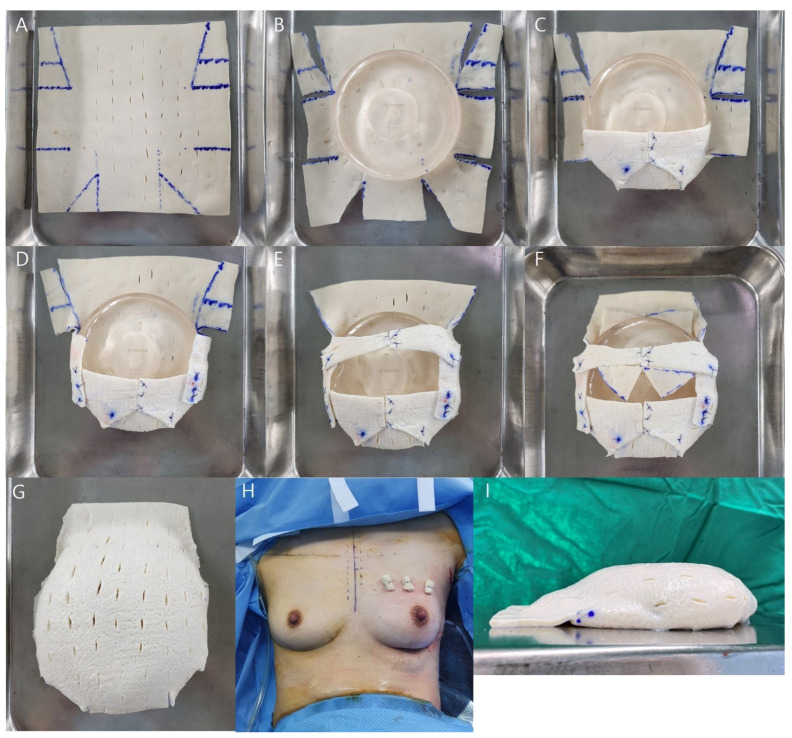
Acellular dermal matrix (ADM) wrapping and surgical procedures in the intraoperative field. (**A**) Wrapping design for ADM. (**B**) ADM cut and implant placement. (**C**) The middle part of the bottom is folded inward first, and both sides are sutured to the center. (**D**) Both middle sides are folded and fixed at the lower pockets. (**E**) Both lateral upper sides are gathered at the center and fixed. (**F**) Two of the cut triangles created from the bottom are sutured to the area not covered by the ADM, and the two created from the top are fixed to the upper ADM for upper pole volume replacement. (**G**) Frontal view of the wrapped ADM. (**H**) Intraoperatively, the implant is fixed to the upper pole with a bolster using the pullout suture technique. This 38-year-old patient was diagnosed with invasive breast cancer of the left breast; the resected breast tissue weighed 224 g. A 170-cc breast implant and CGDerm^®^ 18 × 18 cm^2^ were placed inside. (**I**) Lateral view of the implant wrapped with ADM. The upper pole without implants is filled with ADM to form a natural slope.

**Figure 3 jcm-11-04592-f003:**
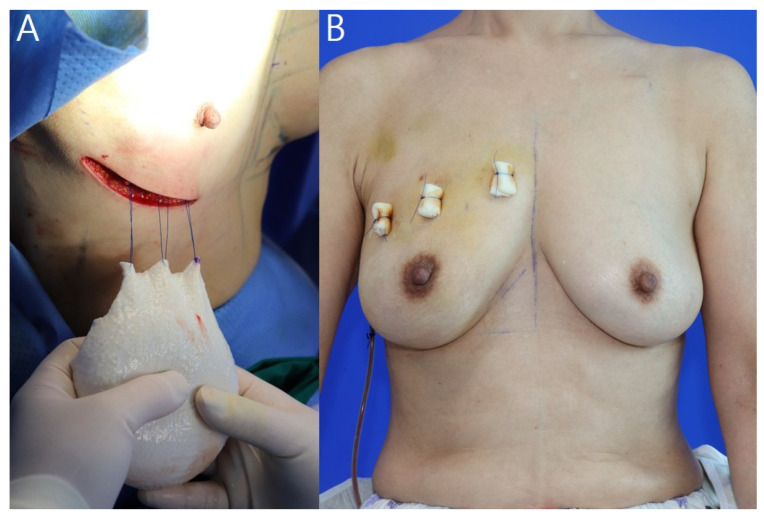
Upper pole pullout suture technique procedures. (**A**) Before inserting the implant, hang it on the ADM upper margin and mastectomy skin upper flap using a Prolene^®^ 2-0 sutures from the outside of the pocket using the pullout suture technique. (**B**) The implant is fixed to the upper pole with a bolster using the pullout suture technique. This 59-year-old patient was diagnosed with invasive breast cancer of the right breast; the resected breast tissue weighed 219 g. A 150-cc breast implant and DermACELL^®^ 16 × 16 cm^2^ were placed inside.

**Figure 4 jcm-11-04592-f004:**
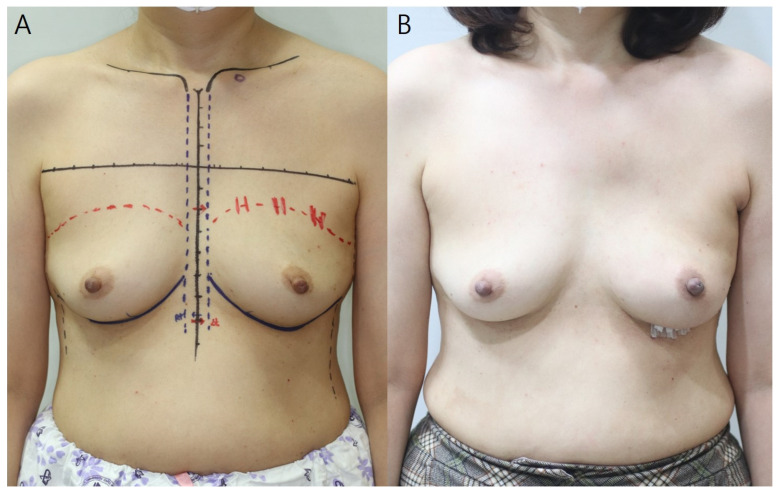
Pre- (**A**) and postoperative (**B**) 12-month photographs. This 54-year-old patient was diagnosed with invasive breast cancer of the left breast; the resected breast tissue weighed 243 g. A 170-cc breast implant and 16 × 16 cm^2^ Megaderm^®^ ADM were placed inside. The red dotted line indicates the upper border of the breast. The area to be anchored to the ADM is marked. ADM, acellular dermal matrix.

**Figure 5 jcm-11-04592-f005:**
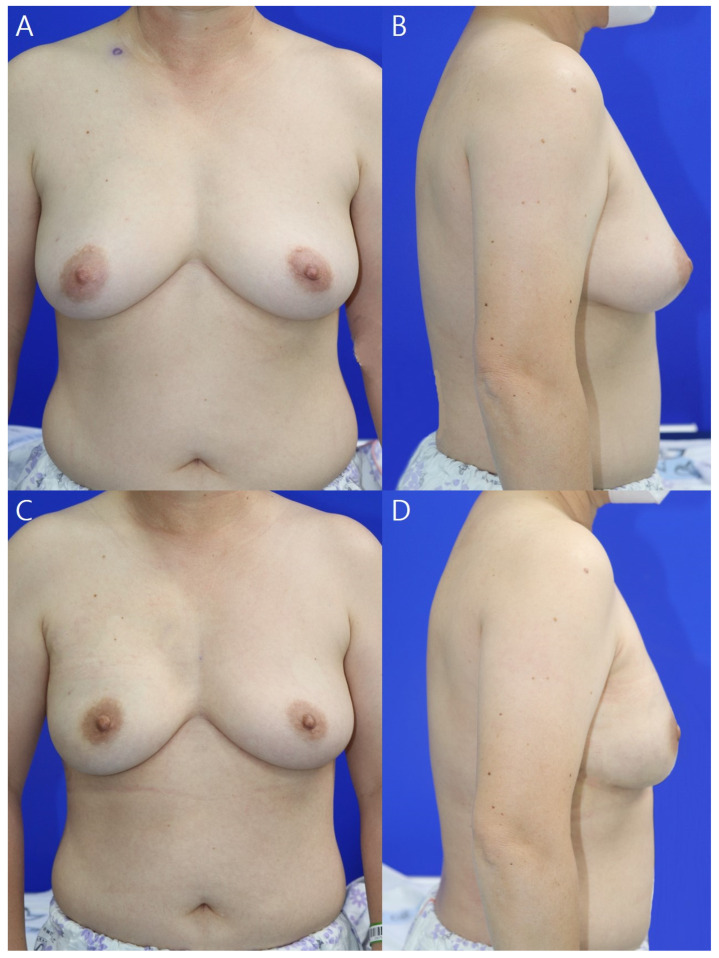
Pre- (**A**,**B**) and postoperative (**C**,**D**) 12-month photographs. This 54-year-old patient was diagnosed with ductal carcinoma in situ of the right breast; the resected breast tissue weighed 334 g. A 275-cc breast implant and 18 × 18 cm^2^ Bellacell^®^ acellular dermal matrix were placed inside.

**Table 1 jcm-11-04592-t001:** Patient baseline characteristics and operative data.

Variables	Value *
Total number of patients	56
Reconstruction side	
Right	30 (53.6%)
Left	26 (46.4%)
Age (years)	50.2 ± 8.9 (32–73)
Weight (kg)	61.3 ± 9.6 (47–102)
Height (cm)	160.1 ± 5.5 (149–170)
BMI (kg/m^2^)	24.0 ± 3.8 (18.1–37.0)
Follow-up period	13.8 ± 1.5 (12–16)
Preoperative breast shape	
Tubular	5 (8.9%)
Round	11 (19.6%)
Conical	19 (33.9%)
Wide	15 (26.9%)
Narrow	6 (10.7%)
Ptotic breasts	27 (48.2%)
Sternal notch to nipple distance (cm)	
Right	22.2 ± 3.1 (14.5–32.5)
Left	22.0 ± 3.0 (15.5–31.0)
Chemotherapy	34 (60.7%)
Neoadjuvant chemotherapy	13 (23.2%)
Adjuvant chemotherapy	21 (37.5%)
Complications	
Seroma	1 (1.8)
Hematoma	1 (1.8)
Infection	0
Mastectomy flap or nipple necrosis	0
Implant malposition, rotation	0
ADM non-incorporation	0

* Values are expressed as median (range) for continuous variables and number (percentage) for categorical variables. Abbreviations: ADM, acellular dermal matrix; BMI, body mass index.

**Table 2 jcm-11-04592-t002:** Patients’ reconstruction details.

Variables	Value *
Breast base width (cm)	13.4 ± 1.8 (10.5–17.5)
Days until JP drain removed	10.4 ± 1.5 (7.0–15.0)
Resected breast tissue weight (g)	274.3 ± 111.4 (77–742)
Mentor silicone implant volume (cc)	230.0 ± 78.4 (130–425)
100–199	23 (41.1%)
200–299	24 (42.9%)
≥300	9 (16.0%)
ADM size (cm^2^)	
16 × 16	26 (46.4%)
18 × 18	23 (41.1%)
20 × 20	7 (12.5%)

* Values are expressed as median (range) for continuous variables and number (percentage) for categorical variables. Abbreviations: ADM, acellular dermal matrix; JP, Jackson–Pratt.

**Table 3 jcm-11-04592-t003:** Satisfactory outcomes using the BREAST-Q and AIS.

	Value (Range)
BREAST-Q score	82.6 ± 12.4 (58–100)
Median AIS scores	
Volume	4.4 ± 0.6 (3.4–4.8)
Shape	4.5 ± 0.6 (3.3–4.8)
Symmetry	4.1 ± 0.8 (2.9–4.3)
Scar	4.1 ± 0.8 (3.1–4.6)
Nipple-areolar complex	4.2 ± 0.6 (3.2–4.6)
Total AIS	21.0 ± 0.3 (19.8–22.6)

Abbreviations: AIS, Aesthetic Item Scale.

## Data Availability

Not applicable.

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
