# Peer review of "“Tear-Drop Appearance” Wrap: A Novel Implant Coverage Method for Creating Natural Contour in Prepectoral Prosthetic-Based Breast Reconstruction"

_jcm, 2022, doi:10.3390/jcm11154592_

Round 1
Reviewer 1 Report
The manuscript is in general well written and the subject is of great general interest among plastic surgeons worldwide.
I have a few comments and remarks.
1. Language: Introduction line 42: the expression “first” is grammatically right but rather informal, so I suggest you replace it by “firstly”. The same goes for “Second” on line 44. Thank you for considering this.
2. Technical point: The description of the surgical technique is thorough, and together with the series of pictures quite understandable. The only thing that needs clarification is exactly how, and when the pull-through stich is put, without risk of breaking the implant. Please add addtionsl figure or more detailed description.
3. Patient selection criteria: You state that the cohort comprises of 56 consecutive patients operated with this method. What is not said, is how many other implant based or autologous breast reconstructions were performed during that same time period, and why some patients were possibly not chosen for this method. What I´m missing here is the “red flag note” do not chose this method for patients with…..
Following on that half the study population are said to have breast ptosis. None of the breasts in the photographs are truly ptotic. Would it be possible to add the sternal notch to nipple measure to Table 1?
4. Discussion and chosen references. When discussing the advantages with PPBR you state that this method eliminates the breast animation deformity (BAD), and use a paper by Pittman from 2018 as a reference. There is however solid research on BAD, showing that BAD may present to some extent also after PPBR . Arch Plast Surg 2019 Nov;46(6):535-543.doi: 10.5999/aps.2019.00493. Epub 2019 Nov 15. Therefore I recommend that you rephrase the discussion line 224 accordingly.
Reviewer 2 Report
The whole structure of this study is good and some corrections are recommended for providing clear information. Particularly, I listed the following comments in detail here.
Major concerns:
In the abstract, the author needs to mention the ingredients of the methods. Also, the finding of the assay could be added step by step based on material and method. I recommend considering regular assays and results. All of the names and terms should be completely mentioned for the first time in text like BREAST-Q.
The introduction is too short in its current form, the authors need to add more information regard such as a breast reconstruction overview and approaches. Then list some disadvantages that cause they to propose “Prepectoral prosthetic-based breast reconstruction (PPBR)”.
Additionally, the citations of the literature are not appropriate, and some sentences lack references such as “Prepectoral prosthetic-based breast reconstruction (PPBR) is a viable option for immediate breast reconstruction after mastectomy”, and so on.
In methods, the author needs to mention the ingredients of the methods and also add a reference to all tests.
In the discussion, discuss your results before relating them to the results of other published work. Also, the author must step by step to come to the results and comparison with others. What is your conclusion? Do the authors have more thoughts on this field?
